# Quantum memory with strong and controllable Rydberg-level interactions

Lin Li[1] & A. Kuzmich[1]

Realization of distributed quantum systems requires fast generation and long-term storage of quantum states. Ground atomic states enable memories with storage times in the range of a minute, however their relatively weak interactions do not allow fast creation of non-classical collective states. Rydberg atomic systems feature fast preparation of singly excited collective states and their efficient mapping into light, but storage times in these approaches have not yet exceeded a few microseconds. Here we demonstrate a system that combines fast quantum state generation and long-term storage. An initially prepared coherent state of an atomic memory is transformed into a non-classical collective atomic state by Rydberg-level interactions in less than a microsecond. By sheltering the quantum state in the ground atomic levels, the storage time is increased by almost two orders of magnitude. This advance opens a door to a number of quantum protocols for scalable generation and distribution of entanglement.

[1] Department of Physics, University of Michigan, Ann Arbor, Michigan 48109, USA. Correspondence and requests for materials should be addressed to A.K. (email: akuzmich@umich.edu).

Atomic systems involving highly excited Rydberg states are an attractive system for the continuing quest to realize large-scale quantum networks[1–6]. An ultra-cold atomic ensemble in a quantum superposition of a ground and a Rydberg state features both rapid and deterministic preparation of quantum states and their efficient transfer into single-photon light fields[7,8]. Notable achievements include the demonstration of deterministic Rydberg single-photon sources[9,10], atom-photon entanglement[11], many-body Rabi oscillations[12–15], photon anti-bunching and interaction-induced phase shifts[16,17] and single-photon switches[18–20]. In parallel to these efforts, significant advances have been made in employing Rydberg interactions for entanglement of pairs of neutral atoms[21–23] and many-body interferometry[24].

All these experimental demonstrations relied critically on the strong interactions between Rydberg atoms. The interactions prevent more than one atom from being excited into a Rydberg state within a volume called the blockade sphere, if excitation into the Rydberg state is slow[7]. In the opposite limit of fast excitation to the Rydberg state, the interactions between the atoms act by dephasing the collective multi-atom states, thereby removing quantum state components with more than one excited atom from the observed Hilbert subspace[25]. Both Rydberg blockade and dephasing mechanisms contribute to the sub-Poissonian statistics of the output light fields in experiments of refs 9,10,12,16.

However, the large values of the electric dipole transition elements between Rydberg states also translate into a magnified sensitivity of these states to black-body radiation and ambient electric fields, leading to their relatively short lifetimes[1,26]. Spontaneous emission, atomic motion and collisions further limit storage times for the ground-Rydberg atomic coherence[9,18]. In contrast, ground atomic states are ideal for preserving quantum coherence[27], but implementation of fast and deterministic quantum operations is challenging due to their weak interactions. For example, deterministic single photons can be produced using measurement and feedback of Raman-scattered light fields[28], but the generation times are $\sim 1$ ms—three orders of magnitude longer than in Rydberg approaches. Such considerations suggest to employ Rydberg levels for interactions and ground levels for storage to achieve both fast quantum operations and long-lived memory. In this work, we demonstrate a quantum memory where a non-classical polariton state created by Rydberg interactions is sheltered in the ground hyperfine sublevels for long-term storage.

## Results

**Experimental set-up.** Our experimental approach is illustrated in Fig. 1: two 795 nm Raman fields ($\Omega_p$ and $\Omega_c$) are applied to create an approximately coherent state of a spin-wave between the ground hyperfine states $|a\rangle$ and $|b\rangle$ in an ultra-cold ensemble of $^{87}$Rb. Next, a 297 nm laser pulse $\Omega_1$ couples state $|b\rangle$ directly to state $|r\rangle$ ($np_{3/2}$) creating a Rydberg polariton state. Subsequently, another 297 nm laser pulse $\Omega_2$ transfers the excitation from the Rydberg state into state $|b\rangle$ for storage. After a storage period $T_g$ in the ground states memory, the read-out field $\Omega_r$ converts the atomic excitation into the retrieved light field. The latter is directed onto a beam splitter and is subsequently detected by single-photon detectors $D_1$ and $D_2$. Additional details of the experimental protocol are given in Supplementary Notes 1 and 2.

**Rydberg excitation.** Single-photon excitation from the ground state $|b\rangle$ to the Rydberg state $|r\rangle$ ($62p_{3/2}$) is studied in Fig. 2. The normalized sum $S_n$ of the $D_1$ and $D_2$ detection rates is shown in Fig. 2a as a function of single-photon detuning $\delta_r$ from the $|b\rangle \leftrightarrow |r\rangle$ resonance. The measured (fwhm) width of the spectrum

$\gamma/2\pi = 1.3$ MHz is largely determined by the 0.7 µs duration of the excitation pulse $\Omega_1$. The population of single excitation prepared in $|b\rangle$, $N$ (at $\delta_r = 0$) is shown in Fig. 2b as a function of Raman excitation population $N_R$ in $|b\rangle$ (no coupling to the Rydberg state). $N$ and $N_R$ are obtained by normalizing the corresponding probabilities of photoelectric detection by the retrieval, transmission and detection efficiencies (Supplementary Note 3). The data are fit with a function of $N = \zeta \chi N_R \exp(-\chi N_R)$, where $\zeta = 0.20(1)$ and $\chi = 0.87(4)$ are adjustable parameters. The fit is suggested by the dephasing mechanism of multi-particle Rydberg excitations put forward in ref. 25. Here $\zeta$ corresponds to the population transfer efficiency of the $|b\rangle \rightarrow |r\rangle \rightarrow |b\rangle$ process in the absence of loss due to multi-particle dephasing, whereas the maximum single excitation preparation efficiency (including multi-particle dephasing loss) in state $|b\rangle$ is $\xi_m = \zeta/e$.

The coherence properties of the ground-Rydberg transition are investigated in Fig. 3a by measuring the retrieved signal as a function of storage time $T_r$ in state $|62p_{3/2}\rangle$. The fast signal decay (with $1/e$ lifetime $\tau_r = 1.58(5)$ µs) is a result of the atomic motional dephasing. During the Rydberg excitation, a spin-wave with phase $e^{i\mathbf{k_1} \cdot \mathbf{r}}$ is imprinted on the ground-Rydberg coherence by the $\Omega_1$ field, where $\mathbf{k_1}$ is the wave-vector of $\Omega_1$, $\mathbf{r}$ is the atomic

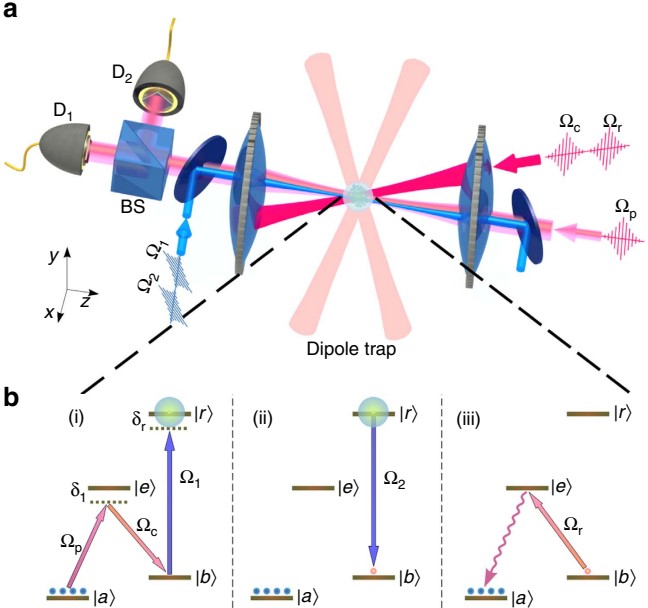

**Figure 1 | Overview of the experiment.** (**a**) Essential elements of the experimental set-up. An ultra-cold $^{87}$Rb gas is confined in a crossed dipole trap formed by two 1,064 nm fields. Two 795 nm beams (probe and control) and a 297 nm beam are focused on the atomic sample with waists ($\omega_p$, $\omega_c$, $\omega_{1,2}$) = (5, 25, 18) µm, respectively. The probe and control beam are aligned with an angle 3°, while the 297 beam counter propagates with the probe beam. Probe $\Omega_p$ and control $\Omega_c$ laser fields are orthogonally circularly polarized. To avoid the dephasing of Rydberg state induced by inhomogeneous light shifts, the dipole trap is turned off before the Rydberg excitation field $\Omega_1$ and switched back on after the Rydberg transfer field $\Omega_2$. (**b**) Level diagram and experimental protocol. (i) Atoms are initially prepared in state $|a\rangle$ by means of optical pumping. The atomic ensemble is driven from $|a\rangle$ to $|b\rangle$ by the probe field $\Omega_p$ and control field $\Omega_c$. Next, the 297 nm field $\Omega_1$ couples $|b\rangle$ directly to the Rydberg state $|r\rangle$, creating a singly excited Rydberg state. (ii) By applying the 297 nm field $\Omega_2$, the short-lived Rydberg excitation is mapped into the ground state $|b\rangle$ for storage. (iii) The ground-state excitation is retrieved by the read field $\Omega_r$ and measured at $D_1$ and $D_2$. The atomic levels involved are $|a\rangle = |5s_{1/2}, F = 1, m_F = 0\rangle$, $|b\rangle = |5s_{1/2}, F = 2, m_F = -2\rangle$, $|e\rangle = |5p_{1/2}, F = 1, m_F = -1\rangle$ and $|r\rangle = |np_{3/2}, m_J = -3/2\rangle$.

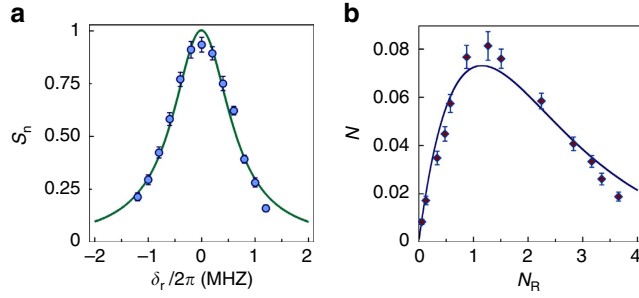

**Figure 2 | Single-photon excitation to Rydberg p state.** (**a**) Single-photon spectroscopy of $|b\rangle \leftrightarrow |r\rangle = |62p_{3/2}, m_J = -3/2\rangle$ transition. The normalized photoelectric detection rate $S_n$ of the retrieved field is shown as a function of detuning ($\delta_r$). The data are fit with a Lorentzian profile. (**b**) $N$, the population of prepared single excitation (with $\Omega_1$ and $\Omega_2$ fields ) is shown as a function of Raman excitation population $N_R$. Error bars, $\pm 1$ s.d.

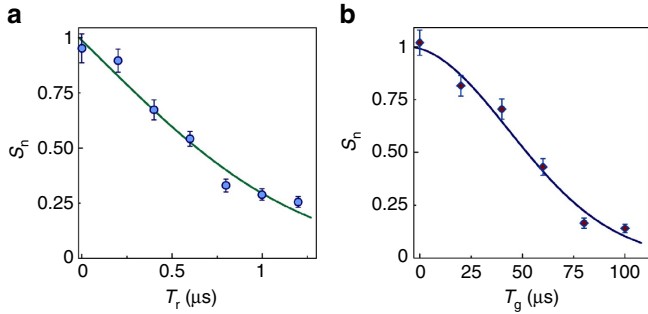

**Figure 3 | Temporal dynamics of atomic polariton.** (**a**) The normalized photoelectric detection rate $S_n$ of the retrieved field is shown as a function of storage time $T_r$ in the Rydberg state. The data are fit with a Gaussian function $\exp(-(T_r + T_d)^2/\tau_r^2)$, while $T_d = 1\,\mu s$ is the delay between two 297 nm fields $\Omega_1$ and $\Omega_2$ for $T_r = 0$ and $\tau_r = 1.58(5)\,\mu s$. (**b**) The normalized photoelectric detection rate $S_n$ of the retrieved field is shown as a function of storage time $T_g$ in the ground states coherence. The data are fit with function $\exp(-(T_g + T_d)^2/\tau_g^2)$, where $T_d = 6\,\mu s$ is the delay between the Raman excitation and the readout for $T_g = 0$ and $\tau_g = 71(2)\,\mu s$. Error bars, $\pm 1$ s.d.

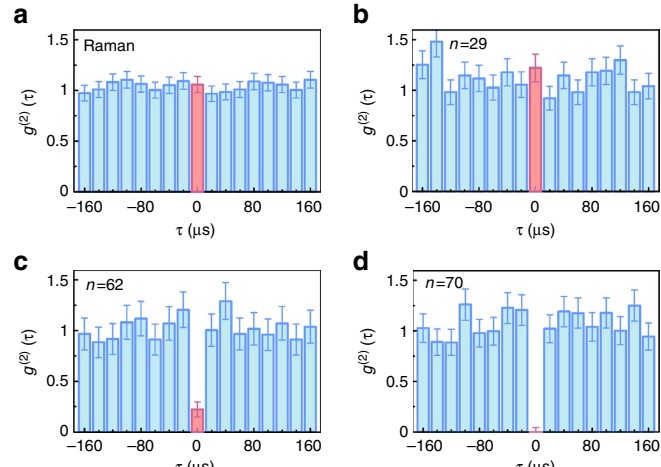

**Figure 4 | Quantum statistics.** Measured second-order intensity correlation function $g^{(2)}$ as a function of delay $\tau$. The data bins for $g^{(2)}(0)$ are highlighted. (**a**) $g^{(2)}(\tau)$ is measured with retrieved coherent light created by the two Raman fields $\Omega_p$ and $\Omega_c$. (**b**–**d**) 297 nm fields ($\Omega_1$ and $\Omega_2$) couple state $|b\rangle$ to a Rydberg state $|np_{3/2}\rangle$, and $g^{(2)}(\tau)$ is measured at $n = 29$, 62 and 70, respectively. Error bars, $\pm 1$ s.d.

position and the spin-wave period is $\Lambda_r = 2\pi/|\mathbf{k_1}| = 297$ nm. For a gas of atoms of mass $M$ at a temperature $T$, atomic motion smears the spin-wave phase grating and leads to a $1/e$ decoherence time of $\tau_r = \Lambda_r/\left(2\pi\sqrt{k_B T/M}\right)$ (refs 28,29), from which the inferred atom temperature is $T \simeq 10\,\mu K$. A lower value of $T \simeq 7\,\mu K$ is found by observing the thermal expansion of the atomic cloud. The difference between the two measurements is a possible indication of atomic heating by the repeated application of the memory protocol. The $\tau_r = 1.58(5)\,\mu s$ coherence time for the $|62p_{3/2}\rangle$ state is nearly identical to the $\tau_r = 1.58(2)\,\mu s$ found for the $|29p_{3/2}\rangle$ state (Supplementary Note 4), indicating the absence of Rydberg interaction-induced decoherence.

**Ground-state coherence.** To achieve longer storage time, we apply the $\Omega_2$ field to coherently transfer the excitation from the Rydberg state $|r\rangle$ to the ground state $|b\rangle$, with the single-photon detuning $\delta_r = 0$. Due to the non-collinear geometry between the probe and control fields with respective wave-vectors $\mathbf{k_p}$ and $\mathbf{k_c}$, the atomic excitation forms a ground states spin-wave, with phase $e^{i\Delta\mathbf{k} \cdot \mathbf{r}}$, where the wave-vector mismatch is $\Delta\mathbf{k} = \mathbf{k_p} - \mathbf{k_c}$ and the spin-wave period is $\Lambda_g = 2\pi/|\Delta\mathbf{k}| = 15\,\mu m$. The stored excitations can be converted into a propagating field by applying a read-out

field $\Omega_r$. To study the temporal dynamics of the quantum memory, the retrieved signal is measured as a function of the storage time $T_g$ in the ground hyperfine sublevels, as shown in Fig. 3b. The observed $1/e$ quantum memory lifetime is $\tau_g = 71(2)\,\mu s$, while the expected lifetime from the scaled value of the ground-Rydberg coherence is $\tau_r \times (\Lambda_g/\Lambda_r) \approx 80\,\mu s$. Assuming the difference in the two values is due to diffusion of atoms out of the ensemble in the transverse ($x$ and $y$) dimensions, we estimate the transverse waist ($1/e^2$) of the atomic ensemble to be $\simeq 6(1)\,\mu m$, which agrees with the measured $5\,\mu m$ waist of the probe field (Supplementary Note 4). In the future, the quantum memory lifetime can be extended into the minute range by employing a suitable state-insensitive optical lattice capable of atom confinement on a length scale smaller than the spin-wave period $\Lambda_g$ (refs 27,30).

**Quantum statistics.** To characterize the non-classical behaviour of our quantum memory, the atomic excitation is read out after a storage time of $T_g = 2\,\mu s$ and a Hanbury Brown-Twiss measurement is performed on the retrieved field with a beam splitter followed by two single-photon detectors $D_1$ and $D_2$. The photoelectric detection events at detectors $D_1$ and $D_2$ are cross correlated, with the resulting second-order intensity correlation function $g^{(2)}(\tau)$ shown in Fig. 4, where $\tau$ is the time delay between the detection events. Panel (a) shows the measurement for an approximately coherent state created by the two Raman fields $\Omega_p$ and $\Omega_c$. The measured second-order intensity correlation function at zero delay $g^{(2)}(0) = 1.06(8)$ is consistent with unity. Panels (b–d) show the quantum statistics of a memory coupled to Rydberg levels $np_{3/2}$ for $n = 29$, 62 and 70, respectively.

As a result of the chosen principal quantum numbers ($n \lesssim 70$) and sample size ($\sim 10\,\mu m$) in our experiment, interactions between the most distant Rydberg atom pairs are in the van der Waals regime, which scale as $\sim n^{11}$ (ref. 1). For low values of $n$, the presence of multiple excitations is expected and the measured $g^{(2)}(0) = 1.22(14)$ for $n = 29$ is consistent with unity. When the interactions are not sufficiently strong for the blockade to be operational over the entire ensemble, more than one Rydberg atom can be excited. Van der Waals interactions lead to the accumulation of phase shifts between different atomic pairs,

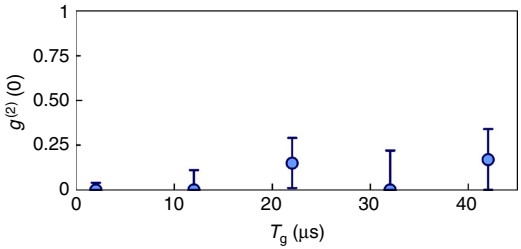

**Figure 5 | Non-classical memory dynamics.** The single excitation generated with the $70p_{3/2}$ state is mapped onto the retrieved field after being stored in the ground states memory for a time of $T_g$. The second-order intensity correlation function at zero delay $g^{(2)}(0)$ is measured at different storage times $T_g$. Error bars, $\pm 1$ s.d.

decoupling them from the phase-matched collective emission mode of the read-out stage. The observed suppression of two-photon events at zero delay for high-lying Rydberg states $n = 62$ and 70 reflects Rydberg excitation blockade and interaction-induced dephasing between multiple excitations and demonstrates the single-photon character of the retrieved field. The transition from the classical statistics to the manifestly quantum regime is associated with an approximately four orders of magnitude increase in the interaction strength from $n = 29$ to 70. The measured values of $g^{(2)}(0) = 0.22(8)$ for $n = 62$ and $g^{(2)}(0) = 0(0.04)$ for $n = 70$ confirm the preparation of single-quanta in the ground memory states. The quantum statistics of the retrieved light field as a function of storage time are shown in Fig. 5, with all the measured values for $g^{(2)}(0)$ well below unity for up to 42 $\mu$s-long storage.

## Discussion

We have demonstrated a quantum memory with 8% efficiency to prepare a single excitation in $<1\,\mu$s, and a memory lifetime of 70 $\mu$s. The storage times can be further extended, conceivably up to and beyond several seconds, by adopting a state-insensitive optical lattice[27,30]. The results presented here show that the two essential quantum network capabilities—fast quantum state generation and long-term storage—can be achieved at the same time in an atomic-ensemble-based system, opening a route toward a broad range of quantum information protocols. In particular, complex quantum states of atomic ensembles can be generated and stored in their ground states and subsequently converted into highly non-classical states of propagating light fields[7].

## Methods

**Preparation of the ultra-cold atomic sample.** To quickly create a dense sample of $^{87}$Rb in a low background pressure environment, a $2D^+$ magneto-optical trap (MOT) is first loaded from the background gas. The 3D MOT is then loaded from the cold atomic beam generated by the $2D^+$ MOT and directed through a differential pumping opening for 300 ms. For the following 22 ms, the gradient of the 3D MOT is increased to 25 G/cm to compress and load the atoms into an optical dipole trap formed by two orthogonally polarized YAG laser beams, intersecting at an angle of 22°. Sub-Doppler cooling of the atoms is performed by increasing the cooling light detuning and decreasing the power of repumper light for 12 ms. The dipole trap beams have a total power of 5 W and transverse waists of 17 and 34 $\mu$m, resulting in a maximum trap depth of $\simeq 560\,\mu$K. The depth of the dipole trap is adiabatically lowered to $\simeq 30\,\mu$K during the 200 ms after the sub-Doppler cooling stage to further cool the atoms, with the atomic temperature of $\sim 7\,\mu$K inferred from the observed rate of thermal expansion. The peak atomic density is $\rho \sim 2 \times 10^{11}\,\text{cm}^{-3}$. The atomic ensemble has $\sim 10\,\mu$m size in the longitudinal ($z$) dimension, while the $\sim 5\,\mu$m waist of the focused probe beam determines transverse ($x-$ and $y-$) dimensions of the ensemble. A bias magnetic field of 3.5 G is switched on and atoms are optically pumped to the $5s_{1/2}$, $F = 1$, $m_F = 0$ state.

**Data acquisition.** In each experimental trial, photoelectric events from detectors $D_1$ and $D_2$ are recorded within a time interval of 200 ns, determined by the length

of the retrieved pulse. The photoelectric detection probability for both detectors is given by $P = P_1 + P_2 = N_1/N_0 + N_2/N_0$, where $N_{1,2}$ are the numbers of detection events recorded by $D_1$ and $D_2$ and $N_0$ is the number of experimental trails. The probability for detecting double coincidences is given by $P_{12}(\tau) = N_{12}(\tau)/N_0$, where $N_{12}(\tau)$ is the number of coincidences from the two detectors with time delay $\tau$. The second-order intensity correlation function is calculated as $g^{(2)}(\tau) = P_{12}(\tau)/(P_1 P_2)$.

**Data availability.** The data that support the findings of this study are available from the corresponding author on request.

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

## Acknowledgements

We thank P. Berman and M. Saffman for discussions, and P. Zhang for experimental assistance. This work was supported by the Atomic Physics Program and the Quantum

Memories MURI of the US Air Force Office of Scientific Research, US Army Research Laboratory and the National Science Foundation.

**Author contributions**

Both authors contributed substantially to all aspects of this work.

**Additional information**

**Competing financial interests:** The authors declare no competing financial interests.

**Publisher's note**: 

