## [Peer Review File · Nature Communications]

Reviewers' comments:

Reviewer #1 (Remarks to the Author):

This paper reports on the realisation of a quantum memory combined with Rydberg atoms allowing, in principle, a combination of strong nonlinear interactions that may be useful for quantum information processing and quantum storage, which is expected to be an essential part of quantum networks and computing.

The result of the experiment is essentially a triggerable single photon source (within the storage window of 70 μ s) at an efficiency of \sim 8%.

On the whole the data is well presented and the results convincing. There are just a few modifications I would suggest before publication:

1) The paper doesn't describe the Rydberg blockade at all, which is the basis of the non-classical state preparation. I think it would be appropriate to include a few sentences on the first page explaining how the blockade leads to single excitations. This will make reading the paper much easier for anyone who is not working directly with Rydberg atoms. This seems appropriate for a general journal such as Nat. Comm. Certainly, as I read it, even though I am quite familiar with the concept, I would have like to see something reminding me how it works.

2) On page one there is a section of text that reads: "When the interactions are not sufficiently strong for the blockade to be operational over the entire ensemble, more than one Rydberg atom can be excited. Van der Waals interactions lead to the accumulation of phase shifts between different atomic pairs, decoupling them from the phase-matched collective emission mode of the read-out stage."

When reading this I was confused. By the time I got to the data in Fig. 4 I understood what was meant, but I might be better to leave the discussion of this effect until there is enough context to make the ideas relevant.

3) I realise that the long-term aim of the work is something beyond a triggered single photon source, but since that is what the work presents, it might be interesting to the reader to have some comparison of the 70 μ s, 8% level of performance to other sources.

Reviewer #2 (Remarks to the Author):

The authors describe a method for fast ($<1 \mu$ s) preparation of quantum memory states, using Rydberg excitations to achieve the strong interactions needed for the fast preparation. The storage time of \sim 70 μ s, while shorter than has been achieved in ground state experiments, is longer than those in Rydberg atomic systems and the authors propose methods for extending the storage time. The research is well-done and well-presented, and it will be of interest to people in the quantum information community. The article is suitable for publication in Nature Communications.

A few minor suggestions:

1) Some more discussion of the single quantum excitation preparation efficiency in the supplemental materials may be helpful, particularly in discussing what factors influence the efficiency and what ways the efficiency could be improved.

2) It may be helpful for clarity to mention in the main text that the atom of interest is Rb-87. It is mentioned in the caption of Fig. 1, but not in the main text.

Authors' reply to the referees regarding manuscript "Quantum memory with strong and controllable interactions"

L. Li and A. Kuzmich

Department of Physics, University of Michigan, Ann Arbor, Michigan 48109

(Dated: August 25, 2016)

We thank both referees for their positive and constructive reviews of our manuscript that has been submitted to *Nature Communications*. We have revised the manuscript in line with their suggestions. Our replies are embedded in their original reports in **boldface**.

I. REFEREE 1

This paper reports on the realisation of a quantum memory combined with Rydberg atoms allowing, in principle, a combination of strong nonlinear interactions that may be useful for quantum information processing and quantum storage, which is expected to be an essential part of quantum networks and computing.

The result of the experiment is essentially a triggerable single photon source (within the storage window of $70 \mu\text{s}$) at an efficiency of $\sim 8\%$.

On the whole the data is well presented and the results convincing. There are just a few modifications I would suggest before publication:

1) The paper doesn't describe the Rydberg blockade at all, which is the basis of the non-classical state preparation. I think it would be appropriate to include a few sentences on the first page explaining how the blockade leads to single excitations. This will make reading the paper much easier for anyone who is not working directly with Rydberg atoms. This seems appropriate for a general journal such as *Nat. Comm.* Certainly, as I read it, even though I am quite familiar with the concept, I would have like to see something reminding me how it works.

We have added/edited the following sentences in the second paragraph of page 1 explaining the role of both excitation blockade and dephasing for the ensemble Rydberg experiments:

All these experimental demonstrations relied critically on the strong interac-

tions between Rydberg atoms. The interactions prevent more than one atom from being excited into a Rydberg state within a volume called the blockade sphere, if excitation into the Rydberg state is slow [12]. In the opposite limit of fast excitation to the Rydberg state, the interactions between the atoms act by dephasing the collective multi-atom states, thereby removing quantum state components with more than one excited atom from the observed Hilbert subspace [26]. Both Rydberg blockade and dephasing mechanisms contribute to the sub-Poissonian statistics of the output light fields in experiments of Refs. [3, 4, 14, 18].

2) On page one there is a section of text that reads: "When the interactions are not sufficiently strong for the blockade to be operational over the entire ensemble, more than one Rydberg atom can be excited. Van der Waals interactions lead to the accumulation of phase shifts between different atomic pairs, decoupling them from the phase-matched collective emission mode of the read-out stage."

When reading this I was confused. By the time I got to the data in Fig. 4 I understood what was meant, but I might be better to leave the discussion of this effect until there is enough context to make the ideas relevant.

We agree with the referee on this and therefore moved the sentence into the description of Figure 4.

3) I realise that the long-term aim of the work is something beyond a triggered single photon source, but since that is what the work presents, it might be interesting to the reader to have some comparison of the $70 \mu\text{s}$, 8% level of performance to other sources.

As a comparison to a Raman scattering ensemble-based single photon source, we added the following sentence into the third paragraph on page 1:

For example, deterministic single photons can be produced using measurement and feedback of Raman-scattered light fields [28], but the generation times are $\sim 1 \text{ ms}$ - three orders of magnitude longer than in Rydberg approaches.

II. REFEREE 2

The authors describe a method for fast ($< 1 \mu\text{s}$) preparation of quantum memory states, using Rydberg excitations to achieve the strong interactions needed for the fast preparation.

The storage time of $\sim 70 \mu\text{s}$, while shorter than has been achieved in ground state experiments, is longer than those in Rydberg atomic systems and the authors propose methods for extending the storage time. The research is well-done and well-presented, and it will be of interest to people in the quantum information community. The article is suitable for publication in Nature Communications.

A few minor suggestions: 1) Some more discussion of the single quantum excitation preparation efficiency in the supplemental materials may be helpful, particularly in discussing what factors influence the efficiency and what ways the efficiency could be improved.

We have added the following material into the discussion of efficiency in the supplemental materials:

For the interaction-induced dephasing mechanism, the efficiency of preparing a retrievable single excitation is limited by $1/e$. By employing Rydberg levels of higher principal quantum number n and/or smaller ensemble volumes, transition into the regime of Rydberg excitation blockade can be achieved, with a corresponding increase in preparation efficiency ξ . The latter is also affected by the (motional) Rydberg-ground decoherence, which can be mitigated by adopting a state-insensitive trap for ground and Rydberg atoms.

2) It may be helpful for clarity to mention in the main text that the atom of interest is Rb-87. It is mentioned in the caption of Fig. 1, but not in the main text.

We have included “ ^{87}Rb ” into the first sentence of the Results section on page 1.

REVIEWERS' COMMENTS:

Reviewer #1 (Remarks to the Author):

The authors have addressed all concerns. I am happy for this to be published.

Reviewer #2 (Remarks to the Author):

My comments in the previous review have been sufficiently addressed and I think the article is acceptable for publication.

REVIEWERS' COMMENTS:

Reviewer #1 (Remarks to the Author):

The authors have addressed all concerns. I am happy for this to be published.

Response: we are glad all concerns have been addressed.

Reviewer #2 (Remarks to the Author):

My comments in the previous review have been sufficiently addressed and I think the article is acceptable for publication.

Response: we are glad all concerns have been addressed.